# Practical and effective diagnosis of animal anthrax in endemic low-resource settings

Olubunmi R. Aminu [1,2]*, Tiziana Lembo [1], Ruth N. Zadoks [1¤], Roman Biek [1], Suzanna Lewis [3], Ireen Kiwelu [4,5], Blandina T. Mmbaga [4,5], Deogratius Mshanga [6], Gabriel Shirima [2], Matt Denwood [7], Taya L. Forde [1]

1 Institute of Biodiversity, Animal Health and Comparative Medicine, University of Glasgow, Glasgow, United Kingdom, 2 Nelson Mandela African Institution of Science and Technology, Arusha, Tanzania, 3 Public Health England, Porton Down, Salisbury, United Kingdom, 4 Kilimanjaro Clinical Research Institute, Kilimanjaro Christian Medical Centre, Moshi, Tanzania, 5 Kilimanjaro Christian Medical University College, Moshi, Tanzania, 6 Tanzania Veterinary Laboratory Agency, Northern Zone, Arusha, Tanzania, 7 Department of Veterinary and Animal Sciences, University of Copenhagen, Copenhagen, Denmark

¤ Current address: Sydney School of Veterinary Science, University of Sydney, Sydney, Australia
* rhoda.aminu@glasgow.ac.uk

**Data Availability Statement:** The data supporting this manuscript are available at http://dx.doi.org/10.5525/gla.researchdata.1057.

## Abstract

Anthrax threatens human and animal health, and people's livelihoods in many rural communities in Africa and Asia. In these areas, anthrax surveillance is challenged by a lack of tools for on-site detection. Furthermore, cultural practices and infrastructure may affect sample availability and quality. Practical yet accurate diagnostic solutions are greatly needed to quantify anthrax impacts. We validated microscopic and molecular methods for the detection of *Bacillus anthracis* in field-collected blood smears and identified alternative samples suitable for anthrax confirmation in the absence of blood smears. We investigated livestock mortalities suspected to be caused by anthrax in northern Tanzania. Field-prepared blood smears (n = 152) were tested by microscopy using four staining techniques as well as polymerase chain reaction (PCR) followed by Bayesian latent class analysis. Median sensitivity (91%, CI $_{95\%}$ [84–96%]) and specificity (99%, CI $_{95\%}$ [96–100%]) of microscopy using azure B were comparable to those of the recommended standard, polychrome methylene blue, PMB (92%, CI $_{95\%}$ [84–97%] and 98%, CI $_{95\%}$ [95–100%], respectively), but azure B is more available and convenient. Other commonly-used stains performed poorly. Blood smears could be obtained for <50% of suspected anthrax cases due to local customs and conditions. However, PCR on DNA extracts from skin, which was almost always available, had high sensitivity and specificity (95%, CI $_{95\%}$ [90–98%] and 95%, CI $_{95\%}$ [87–99%], respectively), even after extended storage at ambient temperature. Azure B microscopy represents an accurate diagnostic test for animal anthrax that can be performed with basic laboratory infrastructure and in the field. When blood smears are unavailable, PCR using skin tissues provides a valuable alternative for confirmation. Our findings lead to a practical diagnostic approach for anthrax in low-resource settings that can support surveillance and control efforts for anthrax-endemic countries globally.

**Funding:** O. R. Aminu was supported by grants from the Bill & Melinda Gates Foundation (Program for Enhancing the Health and Productivity of Livestock, project reference ID 1083453). T. Forde was supported by a Marie Skłodowska-Curie Individual Fellowship (659223), a fellowship from the Natural Sciences and Engineering Research Council of Canada (PDF-471750-2015), and a Biotechnology and Biological Sciences Research Council (BBSRC) Future Leader Fellowship (FORDE/BB/R012075/1). The work was also supported by the Wellcome Trust through a Springboard award (SBF002\1168) to T. Lembo by the Academy of Medical Sciences. The funders had no role in study design, data collection and analysis, decision to publish, or preparation of the manuscript.

**Competing interests:** The authors have declared that no competing interests exist.

## Author summary

Anthrax, an ancient disease largely controlled in the developed world, is still widespread in remote and rural communities of low- and middle-income countries where it affects human and animal health, and livelihoods. To control anthrax effectively, detection and accurate confirmation are important, but solutions need to be feasible for the most-affected areas where resources and infrastructure are typically limited. To achieve this, we assessed a newly proposed stain, azure B, for microscopic confirmation on animal blood smears, as this method can be implemented in low-resource laboratories and in the field. Microscopy using azure B was highly accurate compared to other recommended stains and has the added advantage of being more readily available and convenient. However, blood smear samples were unavailable for more than half of suspected cases. We therefore evaluated a molecular test (PCR) on other sample types–whole blood, blood swabs, skin, and flies–stored at ambient temperature. We show high performance of PCR with skin tissues which were available for 90% of carcasses. Thus, under field conditions, smear samples (when available) and tissue samples are most suitable for diagnostic testing of animal anthrax, whereby microscopy can be conducted in affected areas and PCR in in-country reference laboratories.

## Introduction

Zoonotic diseases have a dual and high burden on the health and livelihoods of people, in addition to their impact on animal health and welfare. The livelihoods of the majority of people living in developing countries depend on livestock farming, and it is estimated that about 80% of households in Africa derive all or part of their income from livestock keeping [1]. Therefore, the control of zoonotic diseases that cause human and animal ill-health–as well as losses to livelihoods–is important and highly relevant to achieving the Sustainable Development Goals [2]. In contrast to emerging diseases, endemic zoonotic diseases like anthrax do not receive the attention needed to control them [3,4]. Anthrax is a bacterial disease caused by *Bacillus anthracis* and primarily affects herbivorous mammals, where it is characterised by sudden deaths in otherwise healthy animals.

In many parts of the developing world, underdiagnosis and misdiagnosis are important reasons for limited availability of data on the prevalence, incidence and impact of endemic diseases [3–5]. Confirmation of anthrax through detection of *B. anthracis* in an animal carcass can be achieved by examination of a stained blood smear, or by culture or polymerase chain reaction (PCR). Culture and (or) PCR are considered to be superior compared to microscopy [6], but require infrastructure and consumables that few laboratories in developing countries have access to. For example, culture must be carried out in laboratory facilities equipped at biosafety level (BSL) 2+ or ideally level 3, which are commonly lacking in areas where anthrax is endemic. Besides the higher costs associated with culture, the occurrence of anthrax in very remote and challenging environments might mean that samples collected are not viable for culture when they eventually reach the laboratory, as *B. anthracis* is easily outcompeted by many other bacterial species [6]. Tests that perform highly on both scientific and convenience criteria are desirable; this is especially true in areas where resources are scarce, and infrastructure is limited. Scientific criteria encompass the ability of a test to distinguish between subjects when the condition under investigation is truly present or absent, while convenience criteria are related to ease or practicality of implementing the test [7]. Convenience and scientific criteria are reflected in WHO's recommendation of ASSURED tests for developing countries (Affordable, Sensitive, Specific, User-friendly, Rapid and robust, Equipment free, and Deliverable to those who need

it) [8]. WHO recommends culture and PCR as the methods of choice for confirmation of anthrax because of their high performance in terms of scientific criteria, but these methods are lacking in convenience criteria, particularly in the context of anthrax-endemic countries. In contrast, smear stain microscopy is a rapid and simple method for detecting *B. anthracis* requiring minimal equipment. It therefore holds great potential value as a field-friendly diagnostic method for anthrax confirmation in low-resource settings. Smear stain microscopy meets many of the ASSURED convenience criteria, however little is known about its performance against scientific criteria for the detection of *B. anthracis* for animal anthrax confirmation.

In 1903, M'Fadyean [9] established capsule staining with polychrome methylene blue (PMB) as a specific technique to detect *B. anthracis* and confirm anthrax. The capsule is a key component of *B. anthracis'* complex surface structure and contributes to the pathogenicity and virulence of the bacterium [10]. The capsule is a specific feature of *B. anthracis* and is not usually produced by closely related bacteria in the same genus such as *B. cereus* and *B. thuringiensis* [11]. Although M'Fadyean PMB staining is the generally-accepted reference standard method for anthrax confirmation by microscopy, quality-controlled PMB has been difficult to obtain commercially since the successful control of anthrax in developed countries [12]. In addition, the stain requires at least 12 months to 'age' in order to develop its metachromatic property, i.e. its ability to distinctly stain the capsule [12,13]. These limitations often prevent the rapid confirmation of anthrax in the field. One of the derivatives of PMB–azure B–has the potential to mitigate the limitations of PMB, as it is readily available and does not require maturation before use. The potential of azure B has been assessed under laboratory conditions on a limited number of smears prepared from isolates of *B. anthracis* from goats and mice [12], but no studies have assessed the stain directly on field samples obtained in endemic settings.

Smear samples are one of the easiest samples to collect and store. This makes them useful where infrastructure for cold chain storage is lacking. However, in areas where anthrax-suspect carcasses may be used by humans for food [14–16] or consumed by scavengers [17,18], smear samples may be difficult to obtain for microscopy. PCR testing using other sample types provides an alternative when laboratory infrastructure is available in-country, although it may not meet some of the ASSURED criteria. For instance, it is more expensive, time consuming and requires more equipment and technical expertise than microscopy. Molecular detection of *B. anthracis* using PCR is a highly specific method to identify the pathogen [19]. However, for animal anthrax, PCR has largely been applied to DNA extracts from *B. anthracis* isolates [19,20], rather than field samples. Confirmation of anthrax from animal tissue samples has been reported using wildlife samples that had been stored in formaldehyde and cryopreserved in liquid nitrogen [21]. There have been no systematic studies to assess both sensitivity and specificity of PCR directly on various sample types, especially those that may be collected under typical field conditions. In addition, it is unknown whether PCR could be conducted on material from slides with stained blood smears, as histological stains may damage *B. anthracis* DNA or affect its integrity or quality, e.g. by intercalating between the genetic material.

Bacterial culture followed by confirmatory tests–including phage and penicillin sensitivity, and PCR to detect genes specific to *B. anthracis*–is currently considered the gold standard approach for the diagnosis of anthrax [6]. Although PCR may detect *B. anthracis* directly from samples where culture has been unsuccessful [22], neither direct PCR, which to our knowledge has rarely been implemented in field settings, nor microscopy are considered gold standard methods for the detection of *B. anthracis*. Therefore, it is important to determine their sensitivity and specificity if they are to be employed to improve the surveillance of anthrax in endemic and resource-poor areas. This study was aimed at testing and providing practical recommendations for the detection of *B. anthracis* from suspect animal carcasses in resource-poor endemic settings where culture is not feasible. To achieve this aim, our study

objectives were to 1) validate the newly proposed azure B staining technique for use on field samples by comparing its sensitivity and specificity to PMB and other routinely used stains; 2) assess the feasibility of using stained and unstained smears in PCR-based anthrax confirmation; and 3) determine the suitability of different sample types for molecular detection of *B. anthracis* using quantitative PCR (qPCR).

## Methods

### Study area

This study was carried out in the Ngorongoro Conservation Area (NCA) of northern Tanzania (Fig 1). The NCA covers an area of 8,292 km$^2$ and had 70,084 inhabitants in 2012, with a population growth rate of 2.7% [23]. The major ethnic group in the study area are the Maasai who practise traditional nomadic pastoralism. The NCA is a multiple-use area where people and animals (including wildlife and livestock) co-exist and it typifies many rural settings in Africa and elsewhere in the world, including the multitude of risks and challenges to the control of neglected diseases. Some of the common characteristics of these settings include the remoteness of communities, the unavailability of well-developed infrastructure, and the co-existence of people and animals. Anthrax is present in the NCA and outbreaks are reported more frequently in the area in comparison to other regions [24,25]. Our ongoing active surveillance (since 2016) in the NCA indicates that the disease is much more widespread in humans and animals than reflected in official reports, with regular cases throughout the year. This provides an ideal setting for assessing the performance of anthrax diagnostic tests.

### Ethics statement

The study received approval from the Kilimanjaro Christian Medical University College Ethics Review committee with certificate No. 2050; National Institute for Medical Research (NIMR), Tanzania, with Reference Number NIMRJHQ/R.8a/Vol. IX/2660; Tanzanian Commission for

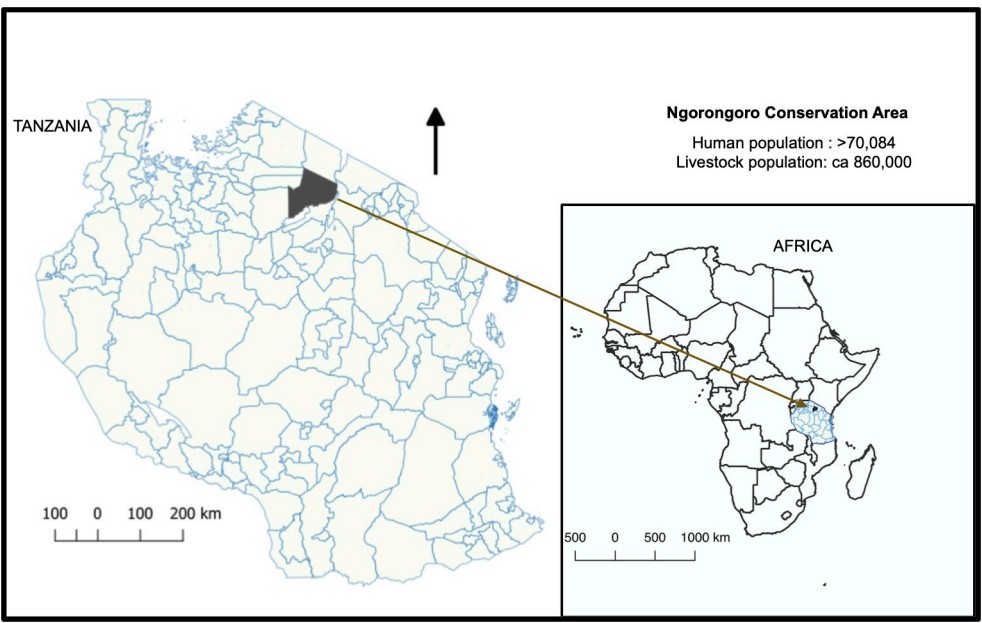

**Fig 1. Tanzania, with the study area for evaluating anthrax diagnostic tools, the Ngorongoro Conservation Area (NCA), shown in grey.** The NCA is a multiple land use area where people, livestock and wildlife live in close proximity and anthrax is endemic. Map was produced using data from Tanzania Bureau of Statistics.

Science and Technology (COSTECH) number 2016-94-NA-2016-88; and College of Medical Veterinary and Life Sciences ethics committee at the University of Glasgow (application number 200150152). Approval and permission to access communities were also obtained from relevant local authorities. Verbal and/or written informed consent was obtained from all owners of livestock sampled after explaining the study objectives. Verbal consent was obtained in lieu of written consent where participants were unable to write. Both verbal and written consent as well as the participant information sheet had been approved by the ethical committees.

## Field-based surveillance and sampling

We set up a field-based active surveillance system within the NCA to investigate deaths in livestock reported by community members suspected to be caused by anthrax, and to obtain samples for disease confirmation based on the identification of *B. anthracis*. Twenty-five local animal health professionals including community animal health workers (CAHWs) and livestock field officers (LFOs) were trained to respond to reports of anthrax cases in the NCA and to collect samples for confirmation. Suspected cases of anthrax in animals were defined as the occurrence of sudden death in previously healthy-looking animals, possibly with associated signs such as blood oozing from the natural orifices and the rapid swelling and decomposition of carcasses. We adapted this definition for wildlife to include carcasses 1) that were sufficiently intact to observe these signs and to preclude death from starvation or predation, and 2) in situations where several wildlife deaths had occurred in an area within a short span of time, suggestive of a disease outbreak. The professionals received sampling kits containing materials for sample collection and personal protective equipment (PPE). Each kit contained primary containers (30 ml Sterilin containers for tissues, soil, flies and swabs, or 5 ml blood tubes for whole blood), secondary containers (Ziplock bags), in addition to slides for blood smear samples, a disposable scalpel and a pair of disposable forceps. The PPE included two pairs of gloves, a face mask, over-sleeves and cover boots, and chlorine release tablets (to be dissolved in water to obtain approximately 10,000 ppm chlorine solution) for decontamination.

Unless otherwise stated, all individual samples were collected into primary containers, then sealed within secondary Ziplock bags. Five sample types were obtained. Firstly, blood smear samples were collected from anthrax-suspect carcasses when blood was available for smearing. In the field at the site of the carcass, blood was smeared onto a slide using a second slide (up to 6 per carcass). Smears were air dried, and slides were carefully wrapped in paper towel and sealed in primary Ziplock bags. Secondly, when available, whole blood was collected using needle-free syringes, transferred into blood tubes with no anticoagulant, wrapped in paper towel and sealed in a secondary container. Thirdly, swab samples were taken by inserting a cotton swab into available blood or fluids from a carcass. Fourthly, depending on the state of the animal remains, tissue was collected from the tip of the ear (if the carcass was still intact) or other available pieces of skin (if they had been butchered or scavenged) (Fig 2). Skin was collected using disposable scalpel and forceps. Finally, flies on and around carcasses and areas where the animals had been butchered or scavenged were collected into tubes. All sample types from a single carcass were packaged in a larger tertiary Ziplock bag and stored at ambient temperature (15˚C to 47˚C in the NCA) prior to transporting to the Kilimanjaro Clinical Research Institute (KCRI) laboratory in Moshi, Tanzania, for testing. Samples were stored at ambient temperature for up to six months before testing.

## Microscopy testing

For each carcass sampled between June 2016 and November 2017 (n = 152), three stains (azure B, Giemsa and Rapi-Diff II) were applied to smear samples (one stain per sample). PMB

a)

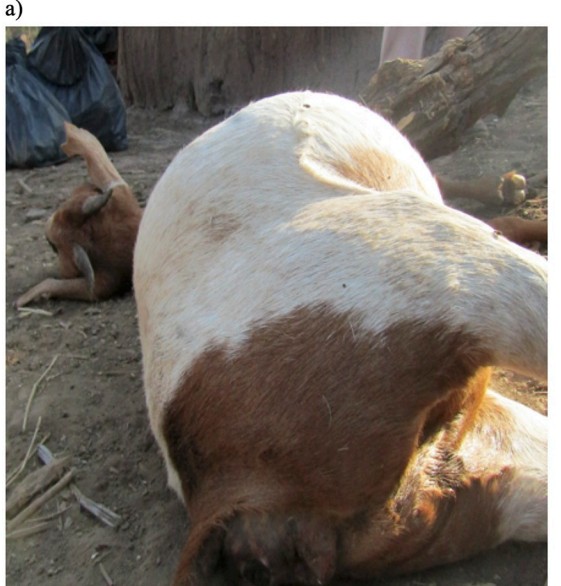

Photo: Rhoda Aminu

b)

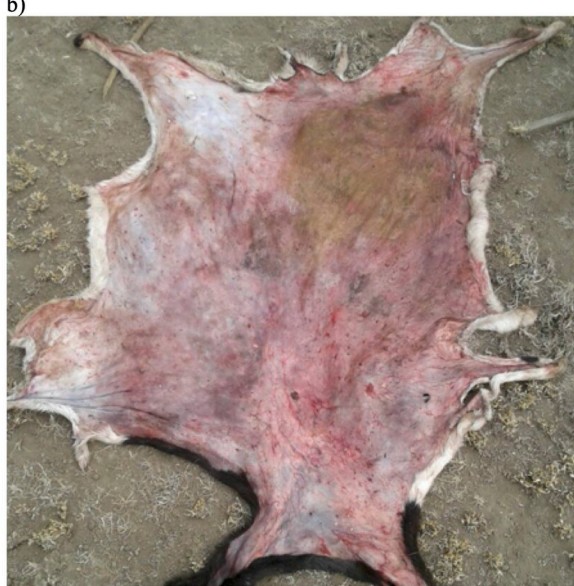

Photo: Sabore Ole Moko

**Fig 2. Examples of suspected *Bacillus anthracis*-infected carcasses sampled as part of a field-based surveillance scheme in northern Tanzania.** Blood smears, whole blood, blood swabs and skin were obtainable from a), whereas only skin could be obtained from b).

staining was carried out on a subset of the total carcasses (n = 102) due to smear sample unavailability. One positive control slide obtained from the Rare and Imported Pathogens Laboratory (RIPL), Public Health England (PHE), was included in each staining batch of up to 12 slides. The control slides consisted of smears of *B. anthracis* isolated from pure culture, fixed in formalin and heat inactivated as per standard procedures carried out by PHE. The staining procedures are outlined in S1 File.

Stained slides were examined using a light microscope (magnification 1000x), in random order with respect to the staining technique. Smears were considered positive if blue or purple square-ended rods were observed surrounded by a pink or pinkish-red capsule or 'shadon', a remnant of capsular material [26] (S1 Fig). A slightly modified protocol, based on [12], was used to define the quality and strength of capsule presence based on the metachromatic property of the stains and the ability to clearly demarcate the capsule from the cells. Scores were assigned to each slide based on the chart shown in S1 Fig.

**Inter-observer and inter-laboratory comparison.** To measure inter-observer variability, which might affect the utility of the test in non-specialist settings, comparisons of slide readings made by multiple observers were carried out. Firstly, a batch of slides stained by one person was viewed and interpreted by two observers. One of the observers, who had previous microscopy experience, but had not been involved in routine *B. anthracis* diagnostics, was briefly trained to identify the morphological characteristics of this pathogen. Observations were carried out on slides that were used to assess the performance of the four stains. Secondly, for azure B only, the two people independently stained and read slides made from the same animal cases (n = 71).

For a subset of suspected anthrax cases, additional blood smear samples (n = 66) were assessed independently by the Tanzania Veterinary Laboratory Agency (TVLA) zonal veterinary centre in Arusha, Tanzania, which is responsible for veterinary diagnostic services within the study region. Here, smears were processed by laboratory personnel following their

routinely used protocol with PMB stain, prepared at the TVLA and aged for 4 years. This procedure is hereafter referred to as the TVLA technique.

## DNA extraction and quantitative PCR testing

All procedures related to sample aliquoting and DNA extraction were carried out in a class 2 biosafety cabinet at a biocontainment level 3 facility at KCRI. Sterile filter pipette tips were used throughout all extractions.

**DNA extraction.** DNA extraction was conducted using the Qiagen DNeasy Blood & Tissue Kit (Qiagen, Germany) spin column protocol, with initial sample preparation conducted as outlined below.

Smear scrapings were collected in a 1.5 ml microcentrifuge tube. After this, 200 μl PBS and 20 μl of 20 mg/ml proteinase K were added to the tubes. For blood samples, 20 μl proteinase K was pipetted into a 1.5 ml microcentrifuge tube. A 100 μl aliquot of the blood sample was transferred into the tube containing proteinase K, and the solution adjusted to 220 μl by adding 100 μl of phosphate buffered saline (PBS). For swabs, the sampled end was cut off and placed into a 1.5 ml microcentrifuge tube and soaked in 200 μl PBS with 20 μl proteinase K. The mixture was incubated at ambient temperature for at least one hour, vortexing the tubes mid-way and after incubation. For skin, a portion (approximately 50 mg) was cut into small pieces of approximately 2 mm$^3$ in a petri dish using a sterile scalpel and transferred into a 2 ml MagNA Lyser bead tube (Roche, United Kingdom). Following this, 360 μl tissue lysis buffer (ATL buffer, included in the Qiagen kit) was added to each tube and the sample was bead beaten four times at 5000 rpm for 18 seconds in a Precellys tissue homogeniser (Bertin, France). Proteinase K (40 μl) was added to the mixture and left to incubate at 56 ˚C for 6 to 8 hours or overnight until complete tissue lysis was achieved. For flies, about 100 mg (between 1 and 3 individuals) were transferred into a 2 ml MagNA Lyser bead tube and 360 μl of PBS was added. The sample was bead beaten four times at 5000 rpm for 18 seconds in a Precellys tissue homogeniser and 200 μl of the homogenised sample was transferred into a microcentrifuge tube with 20 μl of proteinase K added. For all these sample types, the supernatant (220 μl) was transferred to a new microcentrifuge tube and the DNeasy Blood & Tissue Kit spin column protocol was completed according to the manufacturer's protocol. No-template controls were included in each extraction by taking only reagents through the extraction process. All DNA extracts were stored at -20˚C prior to use in PCR. The fly species were not determined prior to DNA extraction.

**qPCR.** Quantitative PCR was carried out on all DNA extracts. Taqman (hydrolysis) probe-based assays were carried out on the Rotor-Gene Q platform (Qiagen), targeting one chromosomal sequence (*PL3*) [19] and two plasmid targets, *cap* (pXO2) and *lef* (pXO1). Primer and probe sequences for the plasmid targets were obtained from RIPL, PHE. Details are available as supplementary materials in S1 Table

Master mix was prepared as follows: 10 μl 2X PrimeTime Gene Expression Master Mix (IDT, Belgium), 10 μM primers and probes (volumes according to S1 Table) and made up to 18 μl per reaction with nuclease free water. The mixture was vortexed and centrifuged briefly. Master mix was added to each qPCR tube and 2 μl of the template DNA was added for a total reaction volume of 20 μl. Negative and positive qPCR controls for each target were included in each run. The cycling conditions were as follows: (1) activation/denaturation at 95˚C for 3 minutes, and (2) amplification, using 40 cycles of 60˚C for 35 seconds and 95˚C for 5 seconds.

**qPCR on DNA extracts from stained smears.** For 15 carcasses testing positive for *B. anthracis* by microscopy and qPCR (based on material from unstained slides), DNA was extracted from each of the 4 stained slides to test whether staining interferes with the qPCR

process. The smear from each slide was scraped off, the DNA was extracted, and qPCR conducted as described above. Primers and probes for only the chromosomal target were used in the qPCR reaction. The cycle threshold (Ct) values for stained smear samples were compared to values for unstained samples. For each stain, 2 PHE controls were included.

### Estimating test sensitivity and specificity

Analyses for estimating the sensitivity and specificity of the tests were conducted assuming the unavailability of a gold standard test by employing latent class analyses [7] within a Bayesian framework. A Latent Class Model (LCM) was applied assuming two latent classes for each of the anthrax-suspect cases studied–anthrax true positive and anthrax true negative carcasses. The LCM formulation that we used is equivalent to an extension of the standard Hui Walter model [27], but our formulation is more similar to that of a state space model where there is a formal separation of the observation layer and the underlying process layer. This type of model can also be considered a generalised form of a mixture model in which the latent classes are related to each other in some way rather than being independent [28]. The analysis used test results from the four different staining techniques and qPCR carried out on blood smear, whole blood, blood swab, and skin tissue samples. A breakdown of the data informing the LCM is shown in Fig 3.

The latent class model estimated the true but latent disease status of each sampled animal as a Bernoulli distribution based on the prevalence of *B. anthracis* within the population of carcasses. This depended on the underlying process (i.e. the presence of capsule or DNA within a sample from an animal infected with *B. anthracis*), which was modelled as a second latent process conditional on the true disease status of the corresponding animal, and the observation process (i.e. the test ability to detect the DNA or capsules in the sample as observed in the PCR

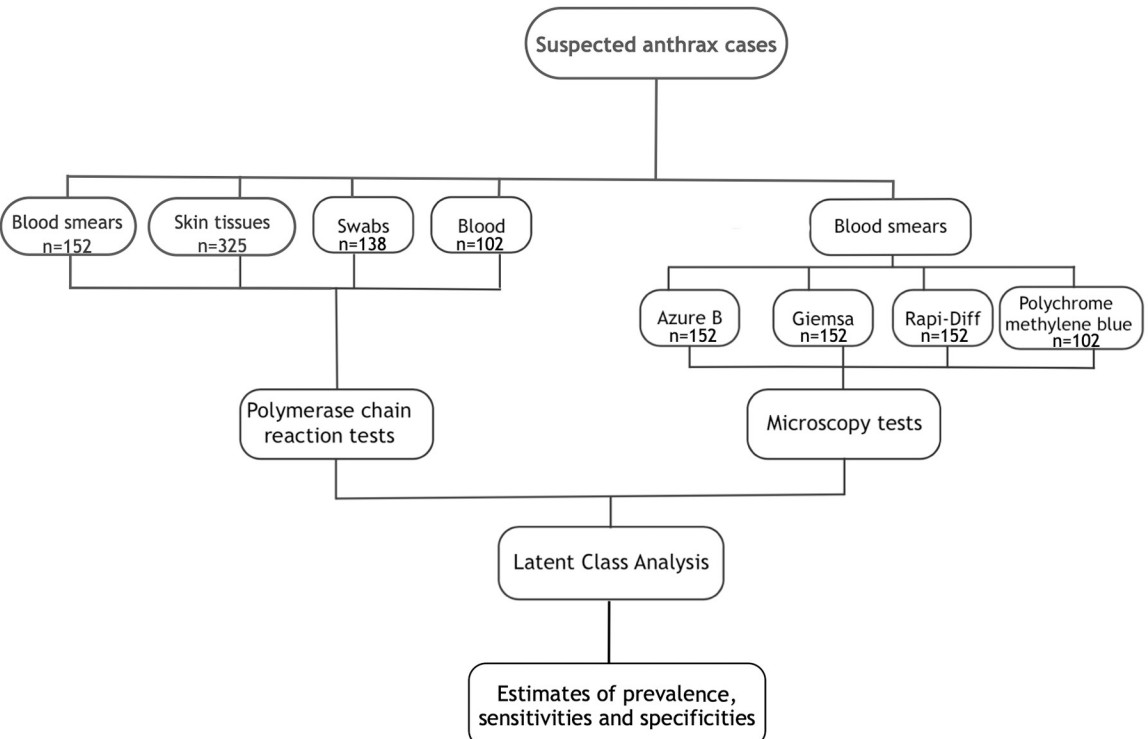

**Fig 3. Workflow for samples and data informing the latent class model used to estimate the sensitivities and specificities of different tests used in the diagnosis of suspected anthrax carcasses.**

and microscopy test results). Minimally informative priors were imposed on the model for the prevalence of *B. anthracis* as no relevant published studies on the prevalence of anthrax (i.e. the proportion of sudden deaths attributable to anthrax) in livestock in Tanzania are available. The prior for prevalence was a Beta (1, 1) distribution. Minimally informative priors were also used for the probability of observing a capsule if present (independent of the stain used), the sensitivities of the four staining techniques, the probability of detecting DNA if present, and the sensitivity of the PCR test. Each of those priors was set to Beta (1, 1).

Much more informative priors were imposed on the model for the specificities of the four staining techniques. This assumed that observing a capsule on a bacillus or chain of bacilli in a sample from a suspected anthrax case was very specific for *B. anthracis*, so the specificities of the test based on this criterion should be high. Thus, Beta (50, 1) indicating specificities between 92% to 100% was used as prior for each of the four staining techniques. For the specificity of PCR, a prior indicating specificity between 92% to 100% (Beta (50, 1)) was also applied to the model. For the underlying processes implying that the presence of the DNA targets and capsule indicate *B. anthracis* infection, priors were Beta (371, 1) and Beta (50, 1), respectively.

The model was fitted using Markov chain Monte Carlo (MCMC) methods implemented using JAGS [29], called from R, version 3.6.0 [30] using the runjags package [31] as an interface. For the model, two MCMC chains each with 20,000 iterations were run. Convergence in the models was assessed visually from the plots generated, as well as from the potential scale reduction factor (psrf) of the Gelman Rubin statistic. Adequate sample size was confirmed using the effective sample size (i.e. $> 400$) of the resulting chains.

Microscopy results as well as qPCR data were treated as binary data, with 0 representing negative results and 1 representing positive results, and were modelled as discrete variables. At the time of the qPCR testing, cut-off Ct values were determined based on the results of the no-template controls. A conservative cut-off value was set at 36 cycles and this cut-off was applied across all three targets to ensure that amplification artefacts such as small-scale cross-contamination or the degradation of probes did not interfere with the qPCR [32]. Samples with Ct values $\leq 36$ were designated positive for the respective target, while those with Ct values $> 36$ or no amplification were considered negative. In the one instance where amplification of a no-template control occurred (Ct value of 37 for target *cap*), the Ct cut-off value was adjusted to 35 for any samples in the same extraction batch. Samples in which all three targets amplified below the cut-off were considered positive for *B. anthracis*. The maximum Ct value of the three values obtained for the different genetic targets was chosen to represent the Ct value for the respective sample. Results of no-template controls were also included in the model, as they provide a form of prior information for the model (true negatives for *B. anthracis*).

The sensitivity and specificity of qPCR were obtained in two ways. In the first, they were derived from the model using a Ct cut-off value of 36. In the second, sensitivity and specificity were estimated by optimising the Ct cut-off. Optimising the balance between sensitivity and specificity yields a threshold for which the total highest sensitivity and specificity are obtained [33].

In assessing the agreement (or disagreement) between different observers, Kappa statistics were used to measure inter-observer agreement and to quantify the consistency of the agreement observed [34]. Kappa statistics for inter-observer agreement, the agreement between the TVLA technique and azure B as well as PMB stain microscopy were computed using the irr package [35] in R version 3.6.0 [30] (S2 File).

## Results

Through the field surveillance platform, 367 suspected anthrax cases were investigated (S2 Table). Blood smears, whole blood, blood swabs and flies were available from 152 (41%), 102

(28%), 138 (38%) and 30 (8%) carcasses, respectively. By contrast, skin samples could be obtained from the vast majority of carcasses (n = 325 or 89%). All five sample types were only obtained from 16 (4%) carcasses.

The majority of cases were sheep (67.3%), followed by goats and cattle (ca. 10% each; S3 Table) and donkeys (4.6%). Non-livestock species included *Giraffa camelopardalis* (giraffe), *Connochaetes taurinus* (wildebeest), *Equus burchellii* (zebra) and *Loxodonta africana* (elephant), while species identity could not be established for 22 suspected cases. The majority of carcasses (80%) had been opened prior to the diagnostic investigation.

## Microscopy

The majority (100/152, 65.8%) of the smear samples were collected less than 24 hours after the death of the animal, while > 98% were collected within a week. The timing of smear collection was not associated with the sensitivity of the microscopy tests, as samples collected more than 24 hours after death were equally likely to be falsely negative as samples collected within 24 hours (for azure B, OR = 1.00, 95% CI (0.98–1.02) (S3 File). The proportion of positives, based on the detection of capsule, was higher with PMB or azure B stains than with Giemsa or Rapi-Diff II (Table 1 and S4 Table).

Microscopy using PMB or azure B had high sensitivity and specificity. In contrast, staining with Giemsa or Rapid Diff II gave poor sensitivity (Table 2).

The LCM allowed the estimation of the sensitivity and specificity of the staining tests as well as the prevalence of anthrax in the samples. The overall prevalence of *B. anthracis* in the samples was estimated to be 68% (95% CI: 62–73%).

Inter-observer agreement was nearly perfect for azure B and PMB (PABAK scores of 0.94 and 0.95, respectively, with 1.0 representing perfect agreement) when both observers evaluated the same slide. Likewise, when different slides were stained and observed separately, inter-observer agreement was near perfect for azure B with a PABAK score of 0.94 (S2 File).

**Table 1. Detection of *Bacillus anthracis* among samples from 152 suspected anthrax cases evaluated with three stains and a subset of 102 cases evaluated with four stains.** PMB = polychrome methylene blue.

| Technique | Number of positive samples | |
|---|---|---|
| | Three stain comparison (n = 152) | Four stain comparison (n = 102) |
| qPCR | 90 (59.2%) | 69 (67.6%) |
| PMB | N/A | 62 (60.8%) |
| Azure B | 81 (53.3%) | 62 (60.8%) |
| Giemsa | 14 (9.2%) | 11 (10.8%) |
| Rapi-Diff II | 15 (9.9%) | 12 (11.8%) |

**Table 2. Estimated sensitivity and specificity of microscopy techniques for detection of *Bacillus anthracis* in blood smears, using a latent class model (LCM), assuming no reference standard.** PMB = polychrome methylene blue.

| Stain | Median sensitivity (posterior 95% credible intervals) | Median specificity (posterior 95% credible intervals) |
|---|---|---|
| Azure B (n = 152) | 90.8% (83.9–96.4%) | 98.5% (96.0–100.0%) |
| PMB (n = 102) | 91.6% (84.3–97.3%) | 98.3% (95.3–100.0%) |
| Giemsa (n = 152) | 16.2% (9.2–24.0%) | 99.2% (97.2–100.0%) |
| Rapi-Diff II (n = 152) | 17.5% (10.3–25.5%) | 99.2% (97.3–100.0%) |

## Quantitative PCR

**Detection of *B. anthracis* in different sample types.** Overall, 61% of samples (457/747) tested positive based on DNA amplification of the three targets at Ct $\leq$ 36. The majority (90%) of samples in which at least one target was detected showed successful amplification of the other two targets as well (Table 3). In many cases, florescence was detected for the *lef* target earliest (39.3% of samples where amplification of all three targets occurred) when compared to the *cap* (6.7%) and chromosomal targets (6.4%). For the other samples (47.6%) Ct values were the same either between two or all three targets. For the *lef* target, samples passed the Ct 1.6 cycles earlier on average compared to both *cap* and the chromosomal target *PL3*.

Quantitative PCR using smear samples yielded the highest combined sensitivity and specificity. For all sample types, the sensitivity and specificity of qPCR were high (87.0% − 98.6%) at the optimal sample-specific threshold (Table 4). The only exception was fly samples, which only had a sensitivity of 19.2%.

**Assessing the possibility of stained smears as starting materials for PCR.** DNA extracts from unstained blood smears from confirmed anthrax cases (n = 15) had lower average Ct values (23.76 +/- 4.85) than those from smears stained with azure B, PMB, Giemsa or Rapi-Diff II (25.82 +/- 4.45, 26.71 +/- 4.81, 28.50 +/- 5.45 and 26.64 +/- 5.01, respectively). Four of the positive controls (one per stain), which had been pre-treated with formalin, showed no amplification.

## Discussion

Our study demonstrates that microscopy using azure B staining on field-prepared blood smears from suspected anthrax-affected animal carcasses from endemic areas yields very high

**Table 3. Number (and percentage) of anthrax-suspected samples with detection of none, one, two, or all three DNA targets at a qPCR cycle threshold ≤36.**

| Sample type | Number of targets amplified | | | |
|---|---|---|---|---|
| | 0 | 1 | 2 | 3 |
| Blood smear (n = 152) | 57 (37.5) | 2 (1.3) | 4 (2.6) | 89 (58.6) |
| Whole blood (n = 102) | 37 (36.3) | 5 (4.9) | 1 (0.1) | 59 (57.8) |
| Blood swab (n = 138) | 43 (31.2) | 5 (3.6) | 7 (5.1) | 83 (60.1) |
| Skin (n = 325) | 82 (25.2) | 15 (4.6) | 5 (1.5) | 223 (68.6) |
| Flies (n = 30) | 20 (66.7) | 4 (13.3) | 3 (10.0) | 3 (10.0) |
| Total samples (n = 747) | 239 (32.0) | 31 (4.1) | 20 (2.7) | 457 (61.2) |

**Table 4. Optimal cycle threshold (Ct) cut-off values and corresponding sensitivity and specificity for detecting *B. anthracis* with quantitative polymerase chain reaction (qPCR) in sample materials from the field and the associated sensitivity and specificity.**

| Sample material | Number of samples available | Median sensitivity at Ct cut-off of 36 (posterior 95% credible intervals) | Median specificity at Ct cut-off of 36 (posterior 95% credible intervals) | Optimal threshold | Median sensitivity at optimal threshold (posterior 95% credible intervals) | Median specificity at optimal threshold (posterior 95% credible intervals) |
|---|---|---|---|---|---|---|
| Blood smear | 152 | 97.8% (93.0–99.7%) | 95.1% (87.0–98.9%) | 32 | 96.2% (90.3–99.2%) | 98.6% (93.2–99.9%) |
| Whole blood | 102 | 87.0% (77.4–93.9%) | 89.3% (77.2–96.2%) | 39 | 93.4% (85.6–97.9%) | 87.0% (74.4–94.9%) |
| Blood swab | 138 | 87.0% (78.6–93.1%) | 93.2% (84.2–98.2%) | 37 | 89.2% (81.3–94.5%) | 92.5% (82.4–98.0%) |
| Skin | 325 | 93.6% (88.9–96.8%) | 94.4% (86.3–98.8%) | 37 | 94.7% (90.2–97.9%) | 94.7% (87.0–98.9%) |
| Flies | 30 | 19.2% (5.1–42.2%) | 93.5% (73.2–99.8%) | 36 | 19.2% (5.1–42.2%) | 93.5% (73.2–99.8%) |

sensitivity and optimal specificity. This technique largely outperforms other stains (Giemsa or Rapi-Diff II) commonly used in laboratories in endemic areas, and is advantageous compared to the gold standard stain–Polychrome Methylene Blue (PMB)–because it can be used immediately after preparation (in comparison to PMB, which requires at least a year of maturation). Azure B staining is also robust, with good consistency of results between users. However, given that in many anthrax-endemic areas carcasses are either consumed or scavenged, blood is often not available for diagnostic confirmation by microscopy. In our study, tissue samples, particularly skin samples, were commonly available from suspect carcasses. We show that these samples enable pathogen detection with high sensitivity and specificity using direct qPCR, even when stored for up to several months at ambient temperature. This sample type can therefore offer a good alternative when microscopy is not possible. These practical solutions will be of considerable value to the surveillance and control of anthrax in other high-risk areas that face similar challenges.

Our results demonstrate that azure B provides a user-friendly alternative to the officially recommended PMB stain for microscopic detection of *B. anthracis*, matching it on scientific criteria (sensitivity, specificity, and inter-observer agreement) and out-performing it on convenience criteria. Smear stain microscopy using azure B fulfils most of the ASSURED criteria. Microscopy is more affordable, user friendly and rapid than culture or PCR; however, deliverability of microscopy under field conditions can be hampered by the limited availability of the officially recommended PMB stain. This limitation is overcome by azure B, which is commercially available and convenient to prepare and use because it does not require aging. Inter-observer agreements for both azure B and PMB indicated that the tests are robust to variability that could occur among multiple observers, and when staining was performed by different individuals–including those with minimal experience–or even laboratories. Thus, azure B is a suitable alternative stain to PMB with major advantages for the detection of *B. anthracis* in blood smear samples from the field. In contrast, the sensitivities of Giemsa and Rapi-Diff II for detecting the capsule of *B. anthracis* were poor and their use should be discouraged for anthrax confirmation.

One major limitation of smear stain microscopy is the need for access to blood samples. In many affected areas in Africa and Asia, anthrax carcasses are consumed by the local population [14–16] or by scavengers [17,18], limiting the availability of fresh samples for diagnostic testing, as confirmed by our findings. This also limits the value of other promising rapid tests, such as lateral flow tests conducted on blood samples [36]. Out of the total number of suspected cases investigated, we could only obtain blood smears for 41%. Therefore, alternative sample materials and diagnostic methods must be considered for anthrax surveillance in endemic areas. Other diagnostic methods include culture and PCR, neither of which would be ASSURED at field level. However, both culture and PCR are considered sensitive and specific, and the requisite facilities and equipment may be available at national level. Because of the need for higher containment facilities when conducting culture of *B. anthracis*, PCR is more user-friendly. Although our work was carried out in a BSL 3 laboratory, this is not a strict requirement for anthrax diagnostic methods. Procedures that do not generate aerosols or large quantities of the pathogen can be safely carried out in lower containment laboratories [6]. In the case of DNA extraction, this would ideally be done within a biosafety cabinet. There are ongoing concerted efforts to build the capacities of national laboratories in endemic countries to carry out molecular detection of *B. anthracis* from clinical specimens [37]. We therefore anticipate that this capacity will be widely available in low-income countries where anthrax is endemic, providing unprecedented opportunities for anthrax surveillance and research. Comparisons between culture and PCR were not conducted as part of the current study due to the lack of local capacity for *B. anthracis* culture. Rather, we focussed on the robustness of PCR to

field conditions (lack of cold chain) and on its ability to deliver results from the available sample types. To our knowledge, this study is the first to demonstrate the value of qPCR, using DNA extracted directly from samples, for the diagnosis of animal anthrax without the need for prior culture.

The convenience criteria we aimed to maximise were those of sample availability, while assessing the suitability of sample storage at ambient temperature to overcome the lack of storage infrastructure in field conditions characteristic of endemic and remote communities in low-income countries. Skin tissue was available in the majority of cases (89%), and from twice as many suspected anthrax cases as blood smears. The high sensitivity and specificity of qPCR using skin samples indicates that the collection of this sample material from suspect animal carcasses has the potential to radically improve anthrax surveillance in endemic settings. PCR using whole blood or swab samples had lower sensitivity and specificity than smear and skin tissue samples, and these sample types suffered from the same challenges of limited availability as described for blood smears; as such, these sample types have minimal utility for routine anthrax surveillance compared to blood smears or skin tissues. While outside the scope of this study, further research would be valuable to assess whether simplified DNA extraction protocols from tissues (i.e. without the need of a homogeniser) would result in similar test accuracy and thereby minimise the amount of specialised equipment required.

We found that it is not only possible to detect *B. anthracis* from samples stored at ambient temperature for up to six months, but that the pathogen can be detected with high sensitivity and specificity. The ability of *B. anthracis* to form spores may be responsible for this observation, since the DNA sequestered in spores is protected from damage. The higher analytical sensitivity of the *lef* target, which is carried by the PXO1 plasmid, is likely associated with the high number of copies of this plasmid typically present in the *B. anthracis* genome. The PXO2 plasmid which carries the *cap* target has been found to be present in much lower copy numbers [38]. No comparisons were made between the outcomes (i.e. sensitivity and specificity) of qPCR for samples stored at ambient temperature and those stored using cold chain. Notwithstanding, the diagnostic results were similar to those obtained using stain microscopy, which is based on a different method of pathogen detection (DNA vs presence of capsule). This suggests that sample storage at ambient temperature is unlikely to have a major impact on the detection of *B. anthracis*.

The poor sensitivity of qPCR for detecting *B. anthracis* from fly samples indicates that they are not useful diagnostic materials, at least in the kind of environments our study was conducted in. The potential to use flies as an indicator of the infection status of associated carcasses depends on the flies being exposed to *B. anthracis* from those carcasses, either through ingestion or as mechanical vectors [39]. Thus, flies not having picked up spores will falsely indicate an infected carcass as negative. The analysis of fly samples has been shown to provide insights into the epidemiology of anthrax in areas where carcasses are even more challenging to find and sample. Hoffmann *et al.* [40] were able to detect the DNA of anthrax-causing *B. cereus* in fly samples, which allowed them to better define the geographical distribution of this pathogen in dense tropical forests in West Africa. DNA was detected in only 5% of the 784 fly samples tested in that study; given the low sensitivity of qPCR we estimated with fly samples, the true prevalence of anthrax could be much higher than what Hoffmann *et al.* reported [40].

Where blood smears can be taken from suspected anthrax cases, microscopy using azure B stained slides should be the method of choice for case confirmation. Furthermore, we found that stained smear samples can also be used reliably for PCR detection. This could be useful in retrospective studies or for molecular investigation of confirmed anthrax cases. However, fixation of blood smears with formalin, which crosslinks and damages DNA, reduced the sensitivity of the qPCR assay, as suggested by previous studies [41]. By contrast, use of azure B, PMB,

Giemsa or Rapi-Diff II does not preclude the use of PCR, although the sensitivity of detection is also slightly reduced. PCR testing on stained smears may not only be useful for the confirmation of anthrax, but may have potential use for strain typing e.g. identifying canonical single nucleotide polymorphisms (SNPs) in the *B. anthracis* genome [42].

## Conclusion

This study, conducted in field conditions in an anthrax-endemic area, has shown that microscopy using azure B in place of PMB is highly sensitive and specific for detecting *B. anthracis* in blood smears from animal carcasses, and more user-friendly because of the availability of azure B. However, tissue samples were more readily available from carcasses than blood smears and *B. anthracis* was detected from them with high sensitivity and specificity using qPCR. In the event of a suspected anthrax case in an animal, smear samples (when available) for use in microscopy and PCR, and skin tissues for PCR are most likely to yield accurate diagnostic results for anthrax surveillance in endemic areas where the lack of infrastructure impedes cold chain storage. We propose practical and feasible solutions to the widely recognised challenges of anthrax surveillance in the most affected areas. Our approaches will therefore be of value to a range of endemic contexts by providing insights into disease occurrence that can be used to inform human and animal health policy and targeted anthrax control efforts.

## Supporting information

**S1 Fig. Chart used to establish presence and strength of *Bacillus anthracis* capsule material.**
(PDF)

**S1 Table. Primer and probe sequences used in the qPCR reactions, targeting two plasmids and one chromosomal sequence in the *B. anthracis* genome.**
(PDF)

**S2 Table. Summary of samples and results for the 367 suspected anthrax cases investigated.**
(PDF)

**S3 Table. Characteristics of the animal carcasses suspected to have died from anthrax, from which samples were collected.**
(PDF)

**S4 Table. Comparison of capsule scores obtained with polychrome methylene blue and azure B.**
(PDF)

**S1 File. Stain preparation and staining procedures.**
(DOCX)

**S2 File. Inter-observer agreement.**
(DOCX)

**S3 File. Effect of sampling time on the sensitivity of smear microscopy.**
(PDF)

## Acknowledgments

We are grateful for all the support received for this research, including from the NCA community. We thank the Ngorongoro District Council (District Veterinary Officer and District

Medical Officer), Ngorongoro Conservation Area Authority (NCAA), Tanzania Wildlife Research Institute (TAWIRI) and members of our field team—Sabore Ole Moko, Sironga Nanjicho, Alutu Masokoto, Godwin Mshumba, Kadogo Lerimba and Paulo Makutian. We are also grateful to the Directorate of Veterinary Services, Ministry of Livestock and Fisheries, and Ministry of Health, Community Development, Gender, Elders and Children. We thank Daniel Bailey of Public Health England, Salisbury for sharing *cap* and *lef* primer sequences. We are grateful to Richard Reeve of the Boyd Orr Centre for Population and Ecosystem Health, University of Glasgow for useful discussions on improving the statistical analysis.

## Author Contributions

**Conceptualization:** Olubunmi R. Aminu, Tiziana Lembo, Ruth N. Zadoks, Roman Biek, Gabriel Shirima, Taya L. Forde.

**Formal analysis:** Olubunmi R. Aminu, Matt Denwood.

**Funding acquisition:** Tiziana Lembo, Gabriel Shirima.

**Investigation:** Olubunmi R. Aminu, Suzanna Lewis, Ireen Kiwelu, Blandina T. Mmbaga, Deogratius Mshanga, Taya L. Forde.

**Methodology:** Olubunmi R. Aminu, Tiziana Lembo, Ruth N. Zadoks, Roman Biek, Matt Denwood, Taya L. Forde.

**Project administration:** Tiziana Lembo, Gabriel Shirima.

**Resources:** Suzanna Lewis, Ireen Kiwelu, Blandina T. Mmbaga.

**Supervision:** Tiziana Lembo, Ruth N. Zadoks, Roman Biek, Gabriel Shirima, Taya L. Forde.

**Writing – original draft:** Olubunmi R. Aminu.

**Writing – review & editing:** Olubunmi R. Aminu, Tiziana Lembo, Ruth N. Zadoks, Roman Biek, Blandina T. Mmbaga, Gabriel Shirima, Matt Denwood, Taya L. Forde.

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
