## [Decision Letter · Decision Letter 0]

9 Jun 2020

Dear Dr. Aminu,

Thank you very much for submitting your manuscript "Practical and effective diagnosis of animal anthrax in endemic low-resource settings" for consideration at PLOS Neglected Tropical Diseases. As with all papers reviewed by the journal, your manuscript was reviewed by members of the editorial board and by several independent reviewers. The reviewers appreciated the attention to an important topic. Based on the reviews, we are likely to accept this manuscript for publication, providing that you modify the manuscript according to the review recommendations. 

Both reviewers raise the question about the limitation of PCR facilities in developing countries - since there was no third reviewer, I also reviewed this manuscript and also had a similar question. I also agreed with the reviewer who suggested that figure S2 might be useful in the main manuscript.

Sincerely,

Brianna R Beechler, Ph.D., DVM

Guest Editor

Ana LTO Nascimento

Deputy Editor

Both reviewers raise the question about the limitation of PCR facilities in developing countries - since there was no third reviewer, I also reviewed this manuscript and also had a similar question. I also agreed with the reviewer who suggested that figure S2 might be useful in the main manuscript.

Reviewer's Responses to Questions

**Key Review Criteria Required for Acceptance?**

**Methods**

-Are the objectives of the study clearly articulated with a clear testable hypothesis stated?

-Is the study design appropriate to address the stated objectives?

-Is the population clearly described and appropriate for the hypothesis being tested?

-Is the sample size sufficient to ensure adequate power to address the hypothesis being tested?

-Were correct statistical analysis used to support conclusions?

-Are there concerns about ethical or regulatory requirements being met?

Reviewer #1: For the purposes of validating the method being investigate here (azure B), it would have been valuable to do culture diagnostics on the same samples. I realize this may not have been feasible in-country, but wonder if it would have been possible to ship the samples out for this analysis?

Reviewer #2: There are no major concerns about the methods.

Just three questions: 

1.) The authors define "suspected cases of anthrax" (lines 182-185). Does this definition also apply for older carcasses where only skin or pieces of skin were available? For livestock, the owners might remember the circumstances of death, but what about wild animals which were found dead and probably widely decomposed? How can anthrax be assumed as cause of death in these cases? 

2.) The scores for staining are indicated in S1 Fig. Which staining method is shown in this Fig?

3.) Did the authors use a balance to determine the weight of the skin portions (50 mg) or was this just estimated?

**Results**

-Does the analysis presented match the analysis plan?

-Are the results clearly and completely presented?

-Are the figures (Tables, Images) of sufficient quality for clarity?

Reviewer #1: (No Response)

Reviewer #2: Results are clearly presented. 

It would be interesting to know which PCR target was the most sensitive. From other publications, one could assume that this would be one of the plasmid targets because plasmids are present in more than one copy.

For completeness, a Table should be added to the supplementary material in which the results for each of the 367 suspected anthrax cases are summarized. This table will show the correlation between the PCR results for different sample materials of the same animal and also between staining and PCR of blood smears.

**Conclusions**

-Are the conclusions supported by the data presented?

-Are the limitations of analysis clearly described?

-Do the authors discuss how these data can be helpful to advance our understanding of the topic under study?

-Is public health relevance addressed?

Reviewer #1: (No Response)

Reviewer #2: The conclusions are supported by the data and limitations were discussed. The public health aspects were well covered. 

Staining and PCR seem to be reliable methods for confirmation of anthrax in low-resource settings. However, DNA extraction using a homogeniser might not be available everywhere. Are there any alternatives for DNA extraction from skin samples without using beads?

**Editorial and Data Presentation Modifications?**

Reviewer #1: L80-83: It seems the problem of lack of infrastructure/consumables would also be a limitation for PCR. May need to add something here to strengthen the argument. Also, it seems the work here was conducted in a BSL3 facility (L248), so it does further raise the question about whether PCR based approaches alleviate the problem. The paper in general could use more discussion on previous use of PCR for anthrax diagnostics in clinical specimens (maybe not here, but in discussion).

L136: Possibly lead here by also indicating what is currently considered to be the “gold standard,” i.e., culture?

L157-158: Would it be possible to qualify the occurrence of anthrax in the NCA? For example, is it widespread and consistent, or does it occur sporadically? This would help support why the NCA may be a good system to validate the diagnostic approaches.

L220: Indicate here that the samples used for PMB staining were a subset of the total samples.

L225, 281, 283: There seems to be a problem with the references here.

L347: Write out “NTC” rather than using acronym, since it only appears a couple times.

L366-367: What proportion of samples had all 5 sample types?

L375-376: Does the time of sampling since death affect smear assay sensitivity? If approximate time since death is known, it may be helpful to assess whether there is a relationship between time since death and probability of being a false negative.

L407-409: This section could use a sentence that highlights which sample type resulted in a qPCR assay with the highest sensitivity/specificity. Also, change “very high” to “high” (L407).

L 446-449: It is not clear what level of observer experience is necessary to achieve the results presented here. This could use some discussion.

L491-500: Perhaps this is stating the obvious, but it may be important to explain why fly samples have low sensitivity. For example, flies found in association with carcasses may not all have been exposed and infected with B. anthracis (so are not true positives), and here you are evaluating their potential as indicator species for the infection status of the nearby carcass.

Table 1: I might suggest renaming the column heading to be “Three stain comparison (n =152)” and “Four stain comparison (n=102)”. But, really, I’m not sure the first column adds that much more, so could be omitted.

Table 2: Should it be 95% credible posterior intervals reported here?

S1 Fig: It would be helpful to include suspect samples that were determined to be negative, so with similar morphology, but lacking the pink capsule or demarcated rod ends.

S2 Fig: I found this Figure helpful to illustrate the approach used, as well as summarize the various methods being assessed. You may want to consider moving this into the main text. Also, would it be possible to breakdown the model into its component classes (e.g., positive/ negative classes)?

Reviewer #2: Some references or links seem to be missing (e.g. line 225)

**Summary and General Comments**

Reviewer #1: This study evaluates the performance of microscopic and molecular-based methods for anthrax diagnostics from field-collected livestock and wildlife carcass samples. The authors’ main objective was to assess the use of a modified blood smear assay (using the azure B stain), which is more accessible in developing countries than culture-based methods. They found that the azure B method had a high sensitivity and specificity in comparison to 3 alternative microscopic staining methods, including the reference standard PMB method. Previous studies have tested the azure B method in the lab (e.g., Owen et al. 2013), but this appears to be the first (?) study to validate its application and performance in the field. Another goal of this study was to evaluate a qPCR-based assay for detecting Bacillus anthracis from a number of carcass sample types. The paper is well written, study design is robust, and conclusions appear to be valid. Ideally, the authors could have included culture data as the gold standard for estimating the sensitivity/specificity of these methods, but it appears there was not a facility in-country equipped for this (hence the need for this paper!). Also, it is not clear whether the study is novel with respect to the use of PCR diagnostics, which is already an accepted standard. Overall, these results provide support for the azure B blood smear method, and has the potential to improve surveillance and disease control in developing countries that lack infrastructure and resources.

Reviewer #2: Anthrax surveillance in endemic low-resource settings is a problem, and the data presented in this manuscript will be helpful to establish reliable methods for confirmation.

PLOS authors have the option to publish the peer review history of their article (what does this mean?). If published, this will include your full peer review and any attached files.

Reviewer #1: No

Reviewer #2: No
---

## [Decision Letter · Decision Letter 1]

28 Jul 2020

Dear Dr. Aminu,

We are pleased to inform you that your manuscript 'Practical and effective diagnosis of animal anthrax in endemic low-resource settings' has been provisionally accepted for publication in PLOS Neglected Tropical Diseases.

Best regards,

Brianna R Beechler, Ph.D., DVM

Guest Editor

Ana LTO Nascimento

Deputy Editor

We thank the authors for completely addressing all the reviewer comments.

Reviewer's Responses to Questions

**Key Review Criteria Required for Acceptance?**

**Methods**

-Are the objectives of the study clearly articulated with a clear testable hypothesis stated?

-Is the study design appropriate to address the stated objectives?

-Is the population clearly described and appropriate for the hypothesis being tested?

-Is the sample size sufficient to ensure adequate power to address the hypothesis being tested?

-Were correct statistical analysis used to support conclusions?

-Are there concerns about ethical or regulatory requirements being met?

Reviewer #2: (No Response)

**Results**

-Does the analysis presented match the analysis plan?

-Are the results clearly and completely presented?

-Are the figures (Tables, Images) of sufficient quality for clarity?

Reviewer #2: (No Response)

**Conclusions**

-Are the conclusions supported by the data presented?

-Are the limitations of analysis clearly described?

-Do the authors discuss how these data can be helpful to advance our understanding of the topic under study?

-Is public health relevance addressed?

Reviewer #2: (No Response)

**Editorial and Data Presentation Modifications?**

Reviewer #2: (No Response)

**Summary and General Comments**

Reviewer #2: (No Response)

PLOS authors have the option to publish the peer review history of their article (what does this mean?). If published, this will include your full peer review and any attached files.

Reviewer #2: No

---

## [Editor Report · Acceptance letter]

4 Sep 2020

Dear Dr Aminu,

We are delighted to inform you that your manuscript, "Practical and effective diagnosis of animal anthrax in endemic low-resource settings," has been formally accepted for publication in PLOS Neglected Tropical Diseases.

Best regards,

Shaden Kamhawi

co-Editor-in-Chief

Paul Brindley

co-Editor-in-Chief
